# Effects of Stepped Heating on the Initial Growth of Oxide Scales on NiCrAlHf Bond Coat Alloy under Air and Water Vapor Atmospheres

**DOI:** 10.3390/ma15082914

**Published:** 2022-04-15

**Authors:** Yang He, Biju Zheng, Peng Song, Taihong Huang, Hezhong Pei, Bixiao Yang

**Affiliations:** Faculty of Materials Science and Engineering, Kunming University of Science and Technology, Kunming 650093, China; hy19940820@163.com (Y.H.); taihonghuang@hotmail.com (T.H.); 15925516074@163.com (H.P.); bixiaoyang@163.com (B.Y.); shakeel07khan@gmail.com (S.)

**Keywords:** oxidation, stepped heating, water vapor, NiCrAlHf bond coat alloy

## Abstract

Temperature and atmosphere have a significant effect on the oxidation of MCrAlY (M = Ni, Co) bond coating. The initial growth behavior of the NiCrAlHf bond coat alloy was investigated at 1100 °C under different atmospheric conditions and using heating methods. A thick Al_2_O_3_ oxide layer and large HfO_2_ particles were observed, perhaps due to metastable oxide growth at low temperatures when using stepped heating. However, in air and water vapor atmospheres, the oxide scale was thinner and the HfO_2_ precipitates were smaller in stepped heating than in constant heating. The size and distribution of the HfO_2_ particles might have induced different microstructures, particularly voids within the oxide scale.

## 1. Introduction

The design of gas turbine burners and the analysis of combustion create fundamental challenges for designers. As we all know, the aero-engine combustion chamber of an aircraft generates high-power work during the take-off process in order to meet the thrust requirements of the take-off. During this process, the combustion chamber employs a rapid heating process, which can be approximately regarded as stepped heating. In order to improve the stability of the combustion chamber of an aero-engine in the later stages of take-off, it is particularly important to study the effects of the use of stepped heating in the early stages on the hot corrosion properties of the materials [1,2].

Ni-based bond coatings are ideal candidates for use as surface protection in the key hot-end components of aero-engines and gas turbines owing to their excellent oxidation resistance properties and thermal stability [3,4]. The performance of an oxide scale depends on the formation of a stable oxide that can protect the alloy from oxidation and corrosion in high-temperature environments. However, in extreme environments, the large consumption of γ-Ni in the material leads to phase transformation in the alloy, whereby β-NiAl/Cr is transformed into γ’-Ni_3_Al/α-Cr to degrade the material [5].

Further, the long-term oxidation behaviors of alloys within the service temperature range have been studied to evaluate their oxidation resistance during service [6,7]. Swadzba et al. [8] reported that long-term oxidation results in the degradation of Al–Si coating after 60 oxidation cycles; however, the coating still provides some protection under cyclic oxidation conditions. Liu et al. [9] prepared a CoCrCuFeNi alloy and their results showed that Cr_2_O_3_, (Cu, Co, Ni)O, and spinel (NiCr_2_O_4_) are formed during long-term oxidation. In the field of high-temperature oxidation, the investigation of the short-term oxidation of alloys is as important as long-term oxidation. Furthermore, after short-term oxidation, Y is distributed as numerous small precipitates within the oxide layer, thus improving the performance of the alloy during thermal cycling [10]. Luo et al. [11] reported that the oxidation of Ni increases the roughness of the surface because the formation of island-shaped oxides accelerates with increasing temperature. Based on previous studies, most short-term oxidations occur at a constant temperature and there have been few reports on the effects of stepped heating on short-term oxidation.

Furthermore, the oxidation resistance of materials has been studied in air and water vapor atmospheres. Lance et al. [12] investigated the oxidation resistance properties of Pt-modified coatings in air and water vapor atmospheres. The results showed that water vapor promotes the formation of the θ-Al_2_O_3_ phase in the surface layer of the coating. Regarding oxidation at a constant temperature, the oxidation resistance of NiAl alloys in a water vapor atmosphere was discussed and it was found that water vapor reduces the densities of the outer oxide layers, meaning that the oxide layer under water vapor conditions is thicker compared to that under air conditions [13]. Further, Wollgarten et al. [14] investigated the influence of water vapor on the oxidation resistance of Ni-based super alloys during short-term oxidation and showed that water vapor promotes the oxidation of Ni and reduces the Ni content of the alloy. In general, water vapor environments reduce the oxidation resistance of materials; however, some studies have shown that they could actually enhance the oxidation resistance of materials. In air and water vapor atmospheres, Hf can improve the adhesion of the oxide scale and extend the life of the material [15]. However, the effects of short-term oxidation on the oxide growth behavior of alloys have rarely been investigated under air and water vapor conditions. In addition, the effects of stepped heating on the growth of alloy oxide scales during the early stages of the oxidation process have also rarely been studied. Therefore, the oxidation mechanisms of Ni-based bond coat alloys in water vapor atmospheres are open for discussion.

At present, most Hf-modified and Ni-based bond coat alloys have been studied for their oxidation resistance at constant temperatures. However, the effects of Hf-modified and Ni-based bond coat alloys on initial oxide scale growth have rarely been studied in water vapor and stepped heating environments. On the basis of the results of previous studies, in this study, we discuss the effects of Hf on the initial growth of oxide scales during stepped and constant heating in air and water vapor atmospheres. We also explain the mechanisms of the effects of HfO_2_ on the oxide scales under different environments.

## 2. Experimental

In this study, NiCrAlHf bond coat alloy (Cr: 15.10 wt.%, Al: 22.50 wt.%, Hf: 0.04 wt.%, balance: Ni) was obtained from electric arc melting, during which the vacuum of the furnace was maintained at approximately 10^−5^ Pa throughout the melting process. The alloy was cut into rectangular (15 mm × 10 mm × 5 mm) blocks using electro-discharge machining. Subsequently, the specimens were polished and ultrasonically cleaned in methanol. The specimens were then placed in an experimental oven and baked for 5 min. Then, they were removed and placed in a vacuum tube furnace for further experiments. Experiments were performed during constant and stepped heating. Specifically, constant heating refers to when a sample was placed in the tube furnace immediately after the furnace temperature reached 1100 °C and stepped heating refers to when a sample was placed in the tube furnace at 25 °C and then heated to 1100 °C in steps. The heating curve is shown in Figure 1 and the service environments and experimental conditions of the specimens are listed in Table 1.

After the oxidation tests, an X-ray diffraction (XRD, D/Max2500PC Rigaku, Tokyo, Japan) analysis was performed using a D8 focusing diffractometer (Bruker, Billerica, MA, USA) in the 2θ range of 20°–90° (λ = 0.15405 nm) to examine the phase composition of the outermost scale of the specimens. A stress analysis of the alloys in different environments was performed using Raman techniques (Renishaw inVia, JCNO, Nanjing, China). The precipitates in the coating were studied using an electron probe microanalyzer (EPMA, JXA-8230), which was equipped with wave-dispersive spectroscopy (WDS), and scanning electron microscopy (SEM, NOVA NANoSEM 450), which was equipped with energy-dispersive spectroscopy (EDS). Further, the precipitates in the coating were also studied using transmission electron microscopy (TEM, JEOL JEM 2100F).

## 3. Results

### 3.1. Microstructures of NiCrAlHf Bond Coat Alloy

The surface morphologies of the alloy at 1100 °C in different environments and using different heating methods are shown in Figure 2. Figure 2a,c shows the surface morphologies after 24 h of oxidation in stepped and constant heating cycles in air, respectively. Figure 2b,d shows the local areas of the surface morphologies of the alloy shown in Figure 2a,c at a high magnification. The SEM images in Figure 2a reveal two surface scale regions with distinct surface morphologies: smooth and rough. The surface characterization of the NiCrAlHf bond coat alloy samples that were heated at high temperatures revealed that the surface region was divided into β-NiAl and γ’-Ni_3_Al phases [16]. Previous studies have found that the oxide scale of Al_2_O_3_ on β-NiAl is thicker than that on the γ’-Ni3Al phase and that the β-NiAl phase is dominated by the aggregation of the Ni–Cr phase. The α-Al_2_O_3_ surface had a typical columnar grain structure, which can also be observed in Figure 2b. In addition, θ-A_2_O_3_ was also formed on the surface. The surface morphologies of the alloy shown in Figure 2c were more uniform compared to those in Figure 2a, and the alumina on the surface of the alloy shown in Figure 2d was denser than that shown in Figure 2b. In addition, the oxide that was generated on the surface of the alloy shown in Figure 2d was α-Al_2_O_3_ and in stepped heating, sparse α-Al_2_O_3_ was formed on the surface of the NiCrAlHf bond coat alloy, which was in contrast to that shown in Figure 2b.

Figure 2e,g shows the surface morphologies after oxidation in stepped and constant heating in a water vapor atmosphere for 24 h, respectively. Figure 2f,h shows the local areas of the surface morphologies of the alloy shown in Figure 2e,g at a high magnification. Figure 2e,g shows that the surface-borne oxides were predominantly strip-shaped Al_2_O_3_ particles, as has also been observed in previous studies. In a water vapor atmosphere, γ-Ni is oxidized to produce NiO, which reacts with Al_2_O_3_ to produce NiAl_2_O_4_ and results in voids on the surface of the alloy [17], as shown in Figure 2e. The sparse distribution of Al_2_O_3_ that was accompanied by the void generation is shown in Figure 2f. Further, the surface morphologies shown in Figure 2g were slightly similar to those in Figure 2a. The surface shown in the SEM image in Figure 2g revealed two surface scale regions with distinct surface morphologies: smooth and rough. The entire surface was divided into β-NiAl and γ’-Ni_3_Al phases. The morphologies of the Al_2_O_3_ particles are shown in Figure 2h as typical bar-like grain structures, and the Al_2_O_3_ particles in Figure 2h were larger than those in Figure 2f. Unlike the air atmosphere, the grain scale of the strip-shaped Al_2_O_3_ particles that were generated by stepped heating in the water vapor atmosphere was much smaller.

### 3.2. Phase Composition and Stress Distribution of NiCrAlHf Bond Coat Alloy

The XRD patterns of the NiCrAlHf bond coat alloy after 24 h of oxidation at 1100 °C in different environments and using different heating methods are shown in Figure 3. The main phases that were present in the alloy mostly comprised Ni_3_Al and Al_2_O_3_, and the XRD results showed that the NiCr_2_O_4_ phase was more easily generated in the water vapor atmosphere than in the air atmosphere. Furthermore, the peak intensity of the NiCr_2_O_4_ phase during constant heating was higher than that during stepped heating in the water vapor atmosphere. In the water vapor atmosphere, a spinel phase (NiAl_2_O_4_) was also observed.

The Raman spectra that were obtained for the NiCrAlHf bond coat alloys in different environments are shown in Figure 4. The peak shifts were predominantly attributed to the large residual internal stresses that were generated inside the alloy at high temperatures. The alumina phase could be determined from the luminescence of Cr^3+^ and the transition from sub-stable to stable alumina at a very early stage has been investigated extensively [18,19]. Tolpygo et al. [20] determined the approximate locations of the α-Al_2_O_3_ and θ-Al_2_O_3_ peaks. The alloy mainly produced α-Al_2_O_3_ and θ-Al_2_O_3_ when using different heating methods and atmospheres. The θ-Al_2_O_3_ peak was more obvious in the water vapor atmosphere, which indicated a high θ-Al_2_O_3_ content and was also consistent with the conclusion that was derived from the surface morphologies of the alloy. In addition, the Raman peak shifted significantly in the environment of water vapor and constant heating, which indicated that the alloy was subjected to a large residual internal stress. Furthermore, the Raman peak also shifted to an extent in the air and stepped heating environment, which indicated that these environments increased the stress generation in the alloy compared to constant heating.

### 3.3. Cross-Sectional Morphologies of NiCrAlHf Bond Coat Alloy

The effects of the different heating methods on the cross-sectional morphologies of the NiCrAlHf bond coat alloy in an air atmosphere are shown in Figure 5. Figure 5a,b shows the cross-sectional morphologies of the alloy that was oxidized for 24 h in an air atmosphere with stepped and constant heating, respectively. Figure 5c,d shows the local areas of the cross-sectional morphologies of the alloy shown in Figure 5a,b at a high magnification. Figure 5a,b shows that the alloy produced a dense and well-bonded oxide scale. The highly magnified images show that stepped heating accelerated the downward segregation of Hf and the formation of large HfO_2_ particles in the oxide scale. Consequently, stepped heating produced a thicker oxide scale than constant heating. Constant heating resulted in a continuous distribution of HfO_2_ in the oxide scale, and HfO_2_ particles that were generated by constant heating were smaller than those generated by stepped heating. A schematic diagram of the oxidation process of the alloy using different heating methods is shown in Figure 5e,f. Stepped heating resulted in a slower transformation of the alumina scale on the alloy; therefore, both θ-Al_2_O_3_ and α-Al_2_O_3_ were formed on its surface, as shown in Figure 5e. Further, the metastable alumina phase caused flimsiness and voids in the alloy and O diffused into the interior of the alloy through the channels provided by those voids, which accelerated the oxidation of the alloy. As O continued to enter, Hf reacted rapidly with O to form HfO_2_ and the continued diffusion of O caused small HfO_2_ particles to aggregate and form larger HfO_2_ particles. In contrast to stepped heating, when the alloy reached 1100 °C in a short time, the aluminum oxide that formed on the surface layer of the alloy under constant heating was α-Al_2_O_3_, as shown in Figure 5f. The stable alumina structure improved the density of the oxide scale on the surface, which could block the entry of O. Therefore, only a small amount of O could enter the interior of the alloy. Thus, the formation of HfO_2_ particles in the oxide scale was more restricted under constant heating than under stepped heating.

The effects of the different heating methods on the cross-sectional morphologies of the NiCrAlHf bond coat alloy in a water vapor atmosphere are shown in Figure 6. Figure 6a,b shows the cross-sectional morphologies of the alloy that was oxidized in a water vapor atmosphere in stepped and constant heating environments for 24 h, respectively. Figure 6c,d shows the local areas of the cross-sectional morphologies of the alloy shown in Figure 6a,b at a high magnification. In the water vapor atmosphere, the downward segregation of the Hf in the oxide scale was significantly higher during constant heating than stepped heating, unlike in the air atmosphere. Figure 6d shows that under water vapor conditions, the oxide scale that was produced during constant heating was sparser and more porous than that produced during stepped heating. In addition, the particle size of the HfO_2_ that was produced by Hf oxidation during constant heating was larger than that produced during stepped heating.

The effects of different the heating methods on the oxidation scale of the NiCrAlHf bond coat alloy in a water vapor atmosphere were then explored further. The oxidation process of the alloy was simulated in a water vapor atmosphere, as shown in Figure 7. A significant amount of the alumina that was on the surface layer of the alloy under water vapor conditions was θ-Al_2_O_3_, which was also confirmed by the surface morphology images that are shown in Figure 1. In addition, the water vapor atmosphere promoted the formation of spinel on the surface of the alloy. The alloy reached 1100 °C within a short amount of time during constant heating. Compared to stepped heating, the alloy remained at 1100 °C under constant heating and the higher temperature promoted the formation of spinel on the surface layer of the alloy. Furthermore, spinel destroyed the structural integrity of the surface layer, which resulted in the alloy becoming flimsy with a porous surface layer. The higher partial pressure of O within the alloy promoted its oxidation. The higher O content promoted the rapid reaction of Hf with O to form HfO_2_ and, unlike during stepped heating, HfO_2_ aggregated to form large HfO_2_ particles that were distributed along the grain boundaries.

## 4. Discussion

### 4.1. Effects of Temperature and Atmosphere on the Transformation of Al_2_O_3_

Based on the aforementioned results, we found that temperature variation had a strong sensitivity to the growth of alumina, as shown in Figure 1. The growth of the larger α-Al_2_O_3_ particles shown in Figure 2b compared to those shown in Figure 2d showed that stepped heating in an air atmosphere accelerated the growth of α-Al_2_O_3_ crystal particles and that the larger particles reduced the density of the surface oxide scale of the alloy to an extent and also reduced the oxidation resistance of the alloy. Previous studies have shown that the minimum temperature at which the θ-Al_2_O_3_ crystalline form starts to transform into α-Al_2_O_3_ in an air environment is 600 °C, while alumina is present in various mixed crystalline forms in the range of 600–950 °C. At 950–1100 °C, alumina transforms into α-Al_2_O_3_ at a faster rate [21,22]; therefore, most of the alumina on the alloy in the air atmosphere was present in the form of α-Al_2_O_3_. Stepped heating reduced the transformation rate of Al_2_O_3_ in the alloy owing to the relatively slow temperature increase compared to constant heating; therefore, both α-Al_2_O_3_ and θ-Al_2_O_3_ were present in the alloy and the oxidation resistance of the alloy was decreased owing to the formation of mixed-phase alumina. In contrast, in a water vapor atmosphere, the alumina was mostly in a θ-Al_2_O_3_ structure, as confirmed in Figure 2f,h. Therefore, stepped heating in the water vapor atmosphere caused the grain scale of θ-Al_2_O_3_ to be significantly smaller than that caused by constant heating. The smaller particle size led to a dense arrangement of alumina on the surface layer of the alloy, which reduced the generation of void and cracks.

During constant heating, the water vapor diffused into the NiCrAlHf bond coat alloy through the grain boundary and reacted with the out-diffusion Al to form AlOOH and H_2_. The specific reaction is shown in Reaction (1) [23]. The generation of H_2_ destroyed the stability of the oxide scale, which resulted in the formation of voids in the oxide scale that then accelerated the oxidation of Al and Hf, as can be observed in Figure 6e. Water vapor reduced the phase transformation rate of Al_2_O_3_ [24]; a large amount of metastable oxide appeared on the surface of the NiCrAlHf bond coat alloy and the metastable oxide reduced the compactness of the surface of the NiCrAlHf bond coat alloy. O diffused into the oxide scale along the grain boundary and reacted with the out-diffusion Al and Hf, which accelerated the oxidation of Al and the nucleation and growth of HfO_2_. Compared to constant heating, the NiCrAlHf bond coat alloy exhibited a thinner oxide layer and smaller HfO_2_ particles during stepped heating, mainly because stepped heating in the water vapor environment inhibited the phase transition of Al_2_O_3_, shrank the volume of Al_2_O_3_, and improved the density of the oxide scale. According to the unpublished research from the laboratory, the specific phase transition temperature and resulting products are shown in Reaction (2).
(1)$$\mathrm{Al}+2H_{2}O=\mathrm{AlOOH}+3/2H_{2}↑$$
(2)$$\gamma -\mathrm{AlOOH}\overset{300-500\;℃}{\to }\gamma \overset{700-800\;℃}{\to }\delta \overset{900-1000\;℃}{\to }\theta \overset{1000-1100\;℃}{\to }\alpha -{\mathrm{Al}}_{2}O_{3}$$

### 4.2. Effects of Heating Method and Atmosphere on the Morphologies of NiCrAlHf Bond Coat Alloy Oxide Scale

To further explore the effects of different atmospheres and heating methods on the initial growth of oxide scale, the area and average thickness of the oxide scale on the cross-sectional NiCrAlHf bond coat alloy were characterized and the results are shown in Figure 8. The results show that in the air atmosphere, the area and average thickness of the oxide scale that was produced during stepped heating were much larger than that produced during constant heating; therefore, stepped heating weakened the oxidation resistance of the alloy in an air atmosphere. Furthermore, in a water vapor atmosphere, the area and average thickness of the oxide scale that was produced during stepped heating were smaller than those produced during constant heating. This also indicated that stepped heating under water vapor conditions improved the oxidation resistance of the alloy, which was consistent with the aforementioned results.

To explore the distribution of Hf in the grains of the alloy, the specimens were cut by FBI and the microscopic morphologies were observed, as shown in Figure 9. From the STEM images and the distribution of Hf at the grain boundaries, we could see that Hf had a larger atomic radius than O. The enrichment of Hf with a larger atomic radius at the grain boundaries could reduce the diffusion path of O to an extent and could also decrease the total energy of the alloy at high temperatures; thus, the energy of the system remained stable and the oxidation resistance of the alloy was improved. Furthermore, the STEM images showed that part of the Hf distribution at the grain boundaries was continuous, which ameliorated the bidirectional diffusion of Al and increased the density of the oxide scale.

As mentioned above, we studied the initial growth of oxide scales on NiCrAlHf bond coat alloy in different atmospheres and using different heating methods to reach 1100 °C. Other studies have shown that the oxidation resistance that is produced by constant heating is better than that produced by stepped heating in an air atmosphere; however, stepped heating is better than constant heating in a water vapor atmosphere. In air, constant heating caused the alumina on the surface of the alloy to become a more stable form of α-Al_2_O_3_. Further, the stable aluminum oxide structure could form a dense oxide scale that provided a good barrier against O. As the temperature gradient changed, the grain scale of Al_2_O_3_ became larger than that produced by constant heating and a mixed crystalline form of alumina occurred on the alloy owing to the fact that stepped heating slows down the transformation of the alumina. Furthermore, the presence of unstable aluminum oxide destabilized the oxide scale on the alloy surface. O could enter the interior of the alloy through voids in the oxide scale, thereby accelerating the oxidation of Hf. As the oxidation went on, small-scale HfO_2_ agglomerated to form large-scale HfO_2_ and segregated in the oxide scale. From previous literature, it has been shown that Hf always segregates in the alumina layer, mainly because Hf reacts with Al_2_O_3_ to form HfO_2_ and the reduced Al continues to be oxidized as the oxidation progresses, then Hf continues to be oxidized. The reaction equation (until the Hf is consumed) is as follows [25]:(3)$$\mathrm{Hf}+O_{2}={\mathrm{HfO}}_{2}\;$$
(4)$${2\mathrm{Al}}_{2}O_{3}{+3\mathrm{Hf}=4\mathrm{Al}+3\mathrm{HfO}}_{2}$$

In a water vapor atmosphere, spinel was produced on the surface layer of the alloy during constant and stepped heating. As constant heating allowed the alloy to reach 1100 °C in a shorter time period, the alloy was more conducive to spinel production. According to previous literature, the production of spinel depletes Al and Cr in the alloy, destabilizes the alloy oxide scale, and creates voids [26]; O enters the alloy through the voids and consequently, the partial pressure of O in the alloy is reduced and the oxidation of Hf is improved.

Furthermore, the distribution of HfO_2_ varied with the different environments and heating methods. In an air atmosphere, stepped heating accelerated the segregation of Hf in the oxide scale and the formation of oxides with O [27]. A portion of singlet Hf also diffused into the alloy along the grain boundaries and increased the bond strength of the oxide scale. Constant heating caused HfO_2_ to form a dense diffusion barrier in the middle of the oxide scale, which hindered the entry of O [28]. In addition, Hf promoted the bidirectional diffusion of Al and improved the density of the oxide scale. In a water vapor atmosphere, Ni reacted with O to form NiO and Ni also reacted with Al_2_O_3_ to form undesirable spinel (NiAl_2_O_4_) and reduced the oxidation resistance of the alloy, which could also be confirmed by XRD. Previous studies have shown that Cr in the alloy forms Cr_2_O_3_ with volatile properties, which provides a channel for the diffusion of O. The volatility of Cr in water vapor environments has also been extensively studied [29,30]. The diffusion of O was significantly faster in the water vapor atmosphere than in the air atmosphere, mainly because H_2_O produces H^+^ and OH^−^ ions through decomposition. H^+^ increases the vacancy concentration in the oxide scale and accelerates the internal diffusion of cations [31,32]. Metal ions can then diffuse into the interior of the alloy with vacancy concentration at the grain boundaries and are accompanied by a certain degree of internal oxidation, which accelerates the growth of oxide scale.

## 5. Conclusions

The effects of different atmospheres and heating methods on the initial growth of oxide scale on a NiCrAlHf bond coat alloy were investigated using XRD analysis, Raman spectroscopy, surface and cross-sectional morphologies, and STEM morphologies. The following conclusions were drawn:

(1)In an air atmosphere, stepped heating slows down the transformation of alumina, reduces the oxygen resistance ability of the alloy surface layer, and accelerates the thickening of the alloy oxide layer. In addition, Hf is rapidly oxidized and large HfO_2_ particles are formed in the oxide scale;(2)In a water vapor atmosphere, constant heating is favorable for the formation of spinel over stepped heating and the formation of spinel can destabilize the alloy and drive oxygen into the alloy, thereby accelerating the thickening of the alloy oxide scale;(3)In a water vapor atmosphere during constant heating, large particles of HfO_2_ can be observed in the oxide scale, which further explains the poor oxygen resistance property produced by constant heating.

## Figures and Tables

**Figure 1 materials-15-02914-f001:**
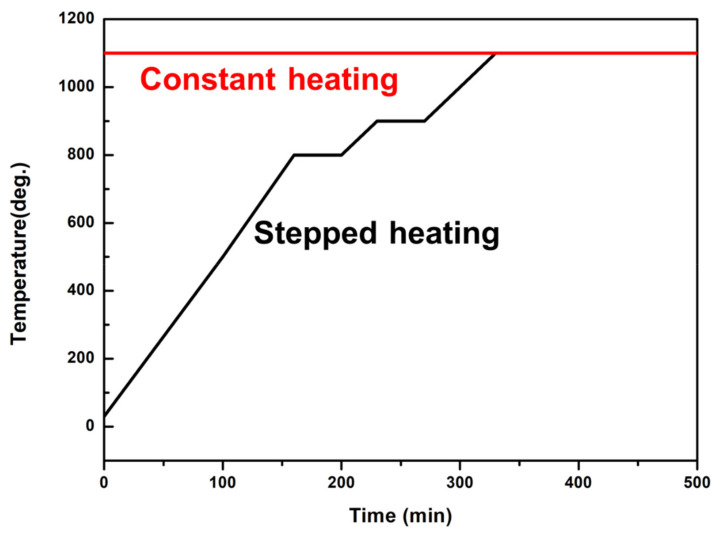
The heating curve of the NiCrAlHf bond coat alloy.

**Figure 2 materials-15-02914-f002:**
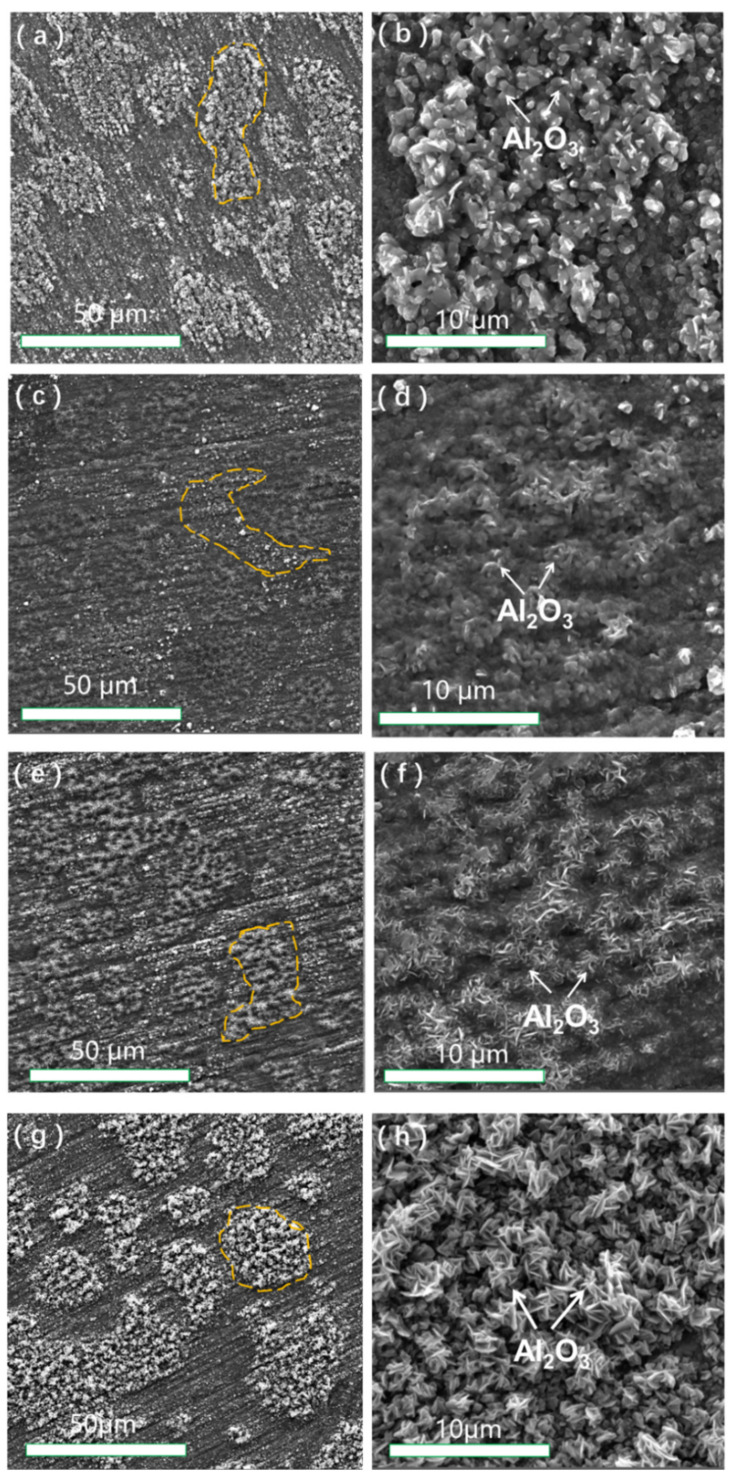
The surface morphologies of the oxide scale formed on the NiCrAlHf bond coat alloy after 24 h of oxidation at 1100 °C: (**a**) in air and stepped heating conditions; (**b**) the high magnification area of (**a**); (**c**) in air and constant heating conditions; (**d**) the high magnification area of (**c**); (**e**) in water vapor and stepped heating conditions; (**f**) the high magnification area of (**e**); (**g**) in water vapor and constant heating conditions; (**h**) the high magnification area of (**g**).

**Figure 3 materials-15-02914-f003:**
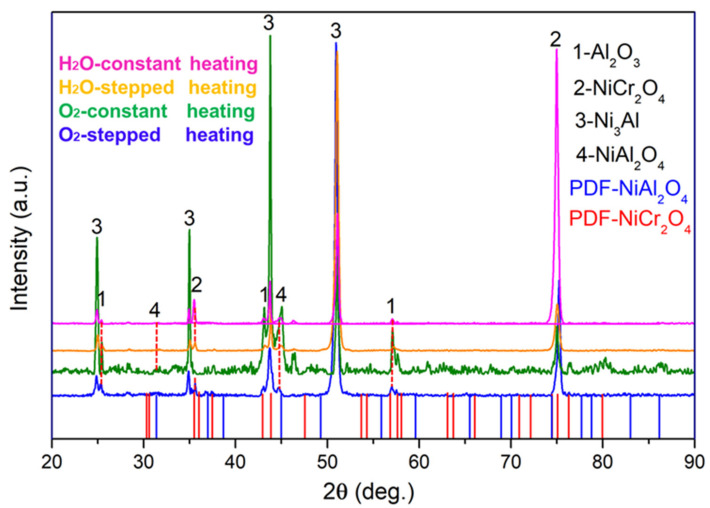
The surface XRD patterns of the oxide scale formed on NiCrAlHf bond coat alloy in different atmospheres and using different heating methods after 24 h of oxidation at 1100 °C.

**Figure 4 materials-15-02914-f004:**
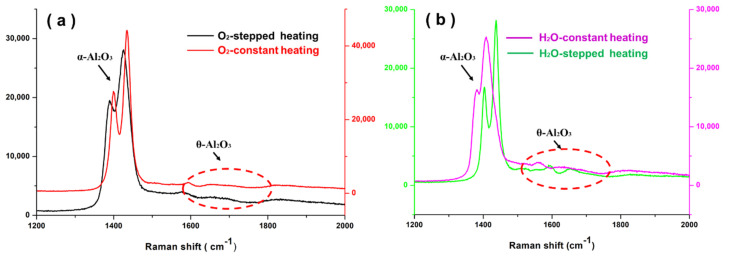
The Raman spectra of the oxide scale formed on NiCrAlHf bond coat alloy in different atmospheres and using different heating methods after 24 h of oxidation at 1100 °C: (**a**) in air; (**b**) in water vapor.

**Figure 5 materials-15-02914-f005:**
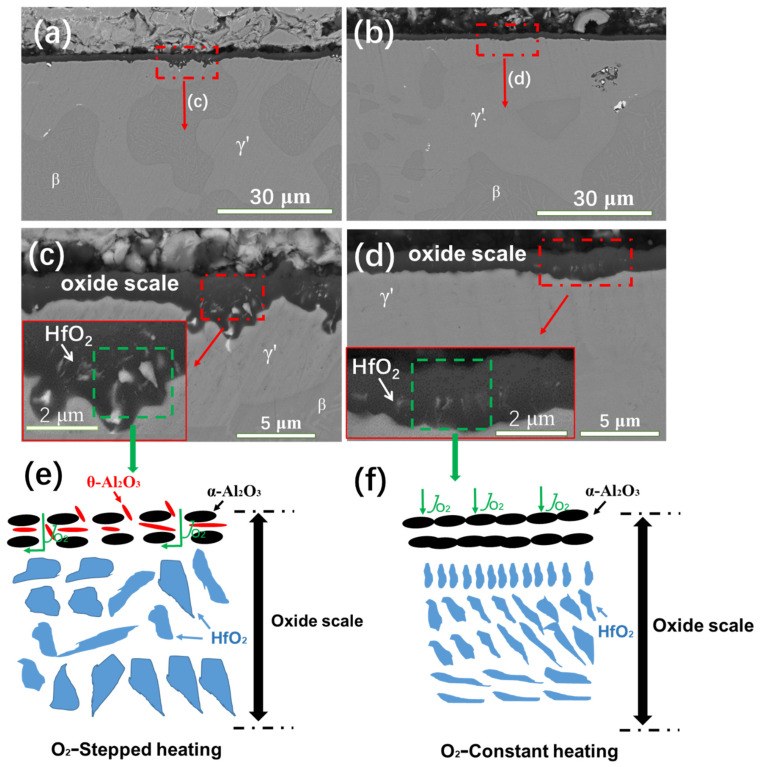
The cross-sectional morphologies of the oxide scale formed on the NiCrAlHf bond coat alloy after 24 h of oxidation at 1100 °C in an air atmosphere: (**a**) during stepped heating; (**b**) during constant heating; (**c**) a local enlargement of (**a**); (**d**) a local enlargement of (**b**); schematic diagrams of the oxidation process of the alloy during (**e**) stepped heating and (**f**) constant heating.

**Figure 6 materials-15-02914-f006:**
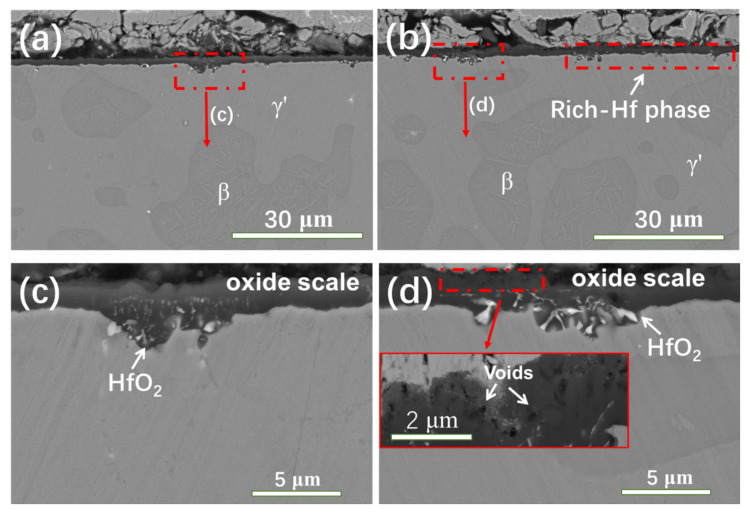
The cross-sectional morphologies of the oxide scale formed on the NiCrAlHf bond coat alloy after 24 h of oxidation at 1100 °C in a water vapor atmosphere during (**a**) stepped heating; (**b**) constant heating; (**c**) the high magnification area of (**a**); (**d**) the high magnification area of (**b**).

**Figure 7 materials-15-02914-f007:**
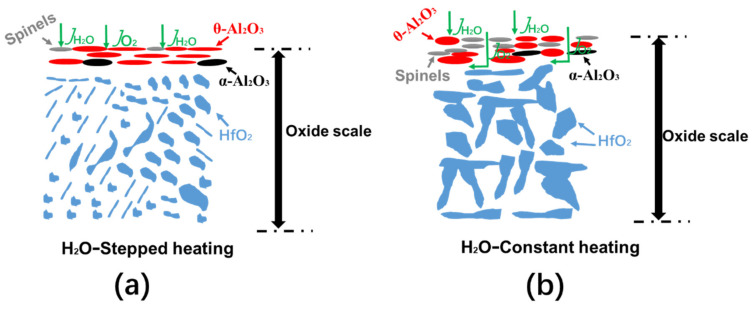
Schematic diagrams of the oxidation process of the NiCrAlHf bond coat alloy in a water vapor atmosphere during (**a**) stepped heating and (**b**) constant heating.

**Figure 8 materials-15-02914-f008:**
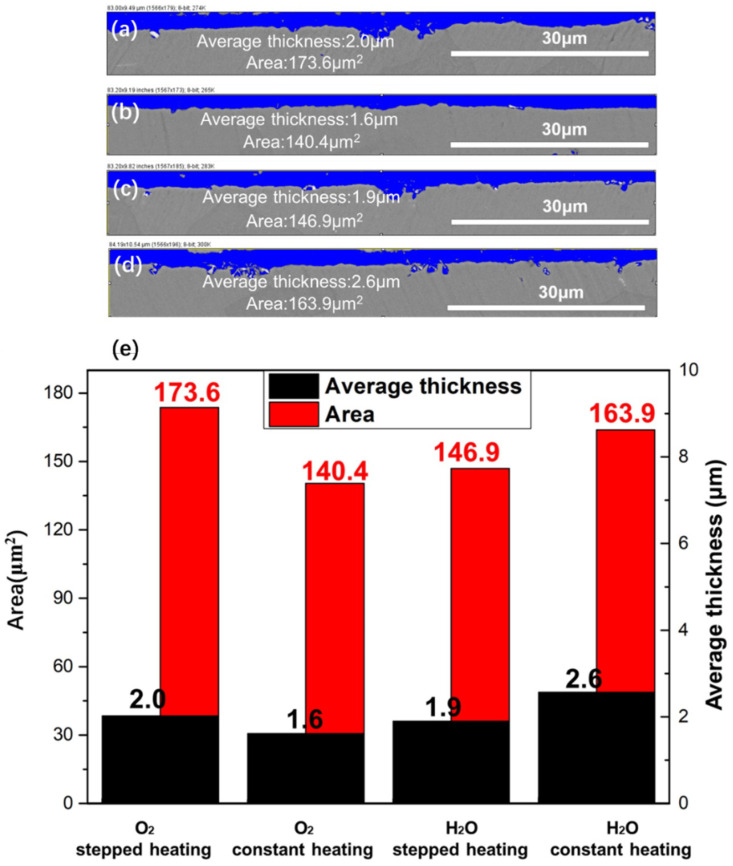
The average thickness and area of the oxide scale formed on the NiCrAlHf bond coat alloy after being oxidized at 1100 °C for 24 h in different atmospheres and using different heating methods: (**a**) in air and stepped heating conditions; (**b**) in air and constant heating conditions; (**c**) in water vapor and stepped heating conditions; (**d**) in water vapor and constant heating conditions; (**e**) a histogram of the results.

**Figure 9 materials-15-02914-f009:**
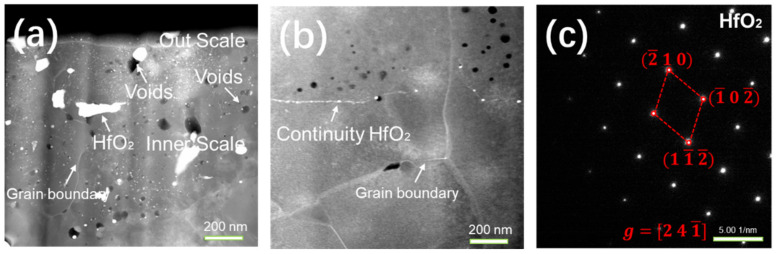
The STEM images of the outer part of the oxide scale formed using different atmospheres and heating methods: (**a**) in air and stepped heating conditions; (**b**) in air and constant heating conditions; (**c**) the SAED patterns of HfO_2_.

**Table 1 materials-15-02914-t001:** The experimental conditions of the alloy in various atmospheres and using different heating methods.

Bond Coat Alloys	Experimental Environment	Experimental Conditions	Holding Time
NiCrAlHf bond coat alloys	Air	Stepped heating/Thermostatic oxidation	24 h
NiCrAlHf bond coat alloys	Air	Thermostatic oxidation	24 h
NiCrAlHf bond coat alloys	Water vapor	Stepped heating/Thermostatic oxidation	24 h
NiCrAlHf bond coat alloys	Water vapor	Thermostatic oxidation	24 h

## Data Availability

The authors declare that all data supporting the findings of this study are available within the article.

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
