# Peer review of "Effects of Stepped Heating on the Initial Growth of Oxide Scales on NiCrAlHf Bond Coat Alloy under Air and Water Vapor Atmospheres"

_materials, 2022, doi:10.3390/ma15082914_

Round 1

Reviewer 1 Report

In this study, the authors studied the oxidation resistance of NiCrAlHf alloy at 1100C in air and water vapour environments. A good combination of experiments were employed to examine the oxide surface and cross-sectional microstructure together with phase composition. However, there are also several major issues that need to be addressed before this paper can be accepted. 

  1. The whole aim of this research seems wrong. The NiCrAlHf alloy developed in this study contains 22.5% of Al and 15.1% of Cr, so it will never be a good candidate for superalloys. It is more suitable for bond coat so it is obvious that the authors did not have a good understanding why they conducted this study. The introduction part should be re-written to promote this as a new concept of bond coat with improved oxidation resistance rather than an alternative superalloys.
  2. It is very rare to see this kind of oxidation test which the authors called it "stepped heating". The reviewer can't think of any real application that requires such a testing condition. Normally we test the bond coats with thermal cycling or iso-thermal oxidation. It is not acceptable unless the authors could provide a good explanation for this special condition. 
  3. Key information is missing in this study. For example, weight gain or TG curve to demonstrate whether Hf indeed improved the oxidation resistance. The authors used thickness measurement from SEM images but it remains unknown whether the TGO scale remain intact during the test or has fallen off. Also the duration of the test is only 24 hours, it is hard to tell which is better. 
  4. The reviewer finds most of the statement in the paper very speculative, e.g. "perhaps", "maybe". If you are not sure, then it is wrong to make vague conclusions especially on the transient oxide formation. 
  5. The authors also failed to provide the detailed testing conditions, i.e. what is the water vapour concentration?
  6. XRD indexing. It is nearly impossible to tell apart NiCr2O4 and NiAl2O4 because both of them are spinel with the same crystal structure. The authors should check their results again and provide more details how these were indexed, i.e. pdf card. 
  7. Raman spectra, theta Al2O3 peak is not visible. It is hard to make any conclusion based on this result.  

Author Response

Thanks to Reviewer 1 for his comments on this manuscript. We have responded to the questions raised by Reviewer 1 one by one. Please refer to the attachment for details.

Reviewer 2 Report

Dear Authors,

I request you to consider the following points,

Kindly revisit the title of the paper.

It is well known that  Temperature and atmosphere have a significant effect on the oxidation of Ni-based superalloys.  What is the reason to mention in the abstract 1st line?

Kindly check the reference style concerning the journal author guidelines.

Table 1: Atmosphere furnace is used. What is the reason to specifically mention about it

Revisit the caption of figures. The present form is very poor.

Figure 2, why there is a 2 magnification. Kindly don’t crop the images as the resolution is very poor. Did the authors absorb any peak shift in the figure 3? If there is no shift, what is the reason/?

The raman shift figure may be inserted in the existing graph and the readers can appreciate it.

Figure 6 is deal with which mode of microscope. Please give detailed information in the captions.

Revisit figure 8.

The language is not good in the present format. Revisit with native English speaker. Furthermore, some of the discussions are very little and the authors mentioned only the available as received datas with poor discussions.

Author Response

Thanks to Reviewer 2 for his comments on this manuscript. We have responded to the questions raised by Reviewer 2 one by one. Please refer to the attachment for details.

Round 2

Reviewer 1 Report

All issues addressed.

Reviewer 2 Report

Dear authors, I have noted the remarkable changes in the manuscript. I will approve the article for further processing through the editor